# Risk of Severe Alphaherpesvirus Infection after Solid Organ Transplantation: A Nationwide Population-Based Cohort Study

**DOI:** 10.3390/biomedicines11020637

**Published:** 2023-02-20

**Authors:** Ya-Wen Chuang, Shih-Ting Huang, I-Kuan Wang, Ying-Chih Lo, Chiz-Tzung Chang, Cheng-Li Lin, Tung-Min Yu, Chi-Yuan Li

**Affiliations:** 1Graduate Institute of Biomedical Sciences, China Medical University, Taichung 404333, Taiwan; 2Department of Post-Baccalaureate Medicine, College of Medicine, National Chung Hsing University, Taichung 40227, Taiwan; 3Division of Nephrology, Department of Internal Medicine, Taichung Veterans General Hospital, Taichung 40705, Taiwan; 4School of Medicine, China Medical University, Taichung 404333, Taiwan; 5Division of Nephrology, China Medical University Hospital, Taichung 404333, Taiwan; 6Division of General Internal Medicine and Primary Care, Department of Medicine, Brigham and Women’s Hospital, Boston, MA 02115, USA; 7Harvard Medical School, Boston, MA 02115, USA; 8Management Office for Health Data, China Medical University Hospital, Taichung 404333, Taiwan; 9Department of Anesthesiology, China Medical University Hospital, Taichung 404333, Taiwan

**Keywords:** solid organ transplant, alphaherpesvirus, herpes simplex virus, herpes zoster, cohort study

## Abstract

Patients after solid organ transplantation (SOT) are more susceptible to various viral infections, including alphaherpesviruses. Therefore, the aim of our study was to investigate the risk of alphaherpesvirus infections, including herpes simplex and herpes zoster, after solid organ transplantation. Inpatient records from the Taiwan National Health Insurance Research Database (NHIRD) defined solid organ recipients, including heart, liver, lung, and kidney, hospitalized for alphaherpesvirus infections as a severe case group of transplants and matched them with a nontransplant cohort. We enrolled 18,064 individuals, of whom 9032 were in each group. A higher risk of severe alphaherpesvirus infection was noted in solid organ recipients (aHR = 9.19; *p* < 0.001) than in the general population. In addition, solid organ transplant recipients had the highest risk of alphaherpesvirus infection within 1 year after transplantation (aHR = 25.18). The comparison found a higher risk of herpes zoster and herpes simplex infections in recipients of kidney (aHR = 9.13; aHR = 12.13), heart (aHR = 14.34; aHR = 18.54), and liver (aHR = 5.90; aHR = 8.28) transplants. Patients who underwent solid organ transplantation had a significantly higher risk of alphaherpesvirus infection than the general population.

## 1. Introduction

Solid organ transplantation (SOT) is considered a major life-saving therapy capable of rescuing seriously ill patients with end-stage organ failure, including heart, lung, liver, and kidney failure. With advances in modern immunosuppressants and perioperative care, short-term outcomes of organ transplantation have improved considerably [1]. Nevertheless, a longitudinal analysis revealed that the long-term outcomes of patients receiving allografts have not improved [2]. Multiple factors are assumed to be involved in disadvantageous posttransplant outcomes, including cancer, diabetes, cardiovascular disease, and overwhelming infections; potent lifelong immunosuppression is suggested to account for this dilemma in recipients of organ transplantation.

Solid organ transplant recipients are susceptible to multiple cutaneous adverse events such as lethal skin cancer, infection, inflammatory dermatitis, etc. The alphherpesviruses (aHV) include three subfamilies: herpes simplex virus (HSV)-1, HSV-2, and varicella zoster virus (VZV). The manifestations of herpes infection constitute a broad spectrum from dermatological lesions to severe visceral organ or neurological complications. In the general population, herpes simplex virus infection is thought to be symptomatically limited, of which HSV-1 infection is occasionally distributed over the orolabial area and HSV-2 is distributed in genital organs. Varicella-zoster virus (VZV), of the human herpesvirus family, is a common viral infection with an estimated seroprevalence rate of over 90% in the majority of global general populations [3]. The risk of herpes virus reactivation increases considerably with advanced age, which is caused by waning cell ability to mediate immunity toward senescence [4,5]. Internal organ involvement is not common in herpes virus infections; however, on some occasions, the reactivation of herpes infection may exacerbate serious conditions, particularly in those who are severely immunocompromised [4,6]. Disseminated VZV infection may exacerbate encephalitis, pneumonia, or hepatitis, contributing to life-threatening circumstances among these patients [5].

A herpes virus test is performed routinely before solid organ transplantation. Patients who have previously been infected with the herpes virus are not barred from receiving organ transplants, so they can receive the transplant regardless of their herpes infection history. Solid organ recipients acquire a lifelong immunocompromised status in preventing allograft organ rejection. The prophylactic protocol for patients undergoing solid organ transplantation caused by aHV infections varies depending on hospital policy. Based on the previous study, SOT recipients have a higher incidence rate than the general population [7]. However, data for evaluating the outcomes of aHV infection in organ transplant recipients remain limited. We designed a national population-based cohort study to examine the risk, timing, and consequences of alphaherpesvirus infection, including HSV and VZV infections, in solid organ transplant recipients.

## 2. Methods

### 2.1. Data Source

This study used data obtained from the National Health Insurance Research Database (NHIRD), which contains the health information of Taiwan’s residents from 1995 until now. This comprehensive database contains outpatient diagnoses, hospitalization (inpatient) records, and the prescription information of the population. For privacy protection, the data were deidentified, recoded, or encrypted. The diagnoses were coded according to the International Classification of Disease, 9th Revision, Clinical Modification (ICD-9-CM). The Research Ethics Committee of China Medical University and Hospital in Taiwan approved this study (CMUH104-REC2-115-AR4).

### 2.2. Study Population

To investigate the association between SOT and aHV, we defined patients who received solid organ transplantation (ICD-9 CM: 996.81, 996.82, 996.83, 996.84, 996.86, V420, V421, V426, and V427,) during 2000–2012 as the case cohort and defined the date that the recipients accepted the SOT as the index date by retrieving hospitalization medical records (inpatient). We defined the comparison group as patients without a history of SOT (non-SOT). The control group was established via 1:1 propensity score matching with the case group according to age, gender, index year, and associated comorbidities. All patients included were older than 18 years. The protocols for managing patients with solid organ transplants caused by aHV infections included routinely administering intravenous or oral acyclovir or other antiviral medications.

The primary outcome in this study was VZV infection (ICD-9-CM: 053) and HSV infection (ICD-9-CM: 054). The covariates included a history of hypertension (ICD-9-CM: 401–405), hyperlipidemia (ICD-9-CM: 272), diabetes mellitus (ICD-9-CM: 250), cancer (ICD-9-CM:140–208), chronic kidney disease (ICD-9-CM: 580–589), chronic obstructive pulmonary disease (COPD; ICD-9-CM: 491, 492, 496), heart failure (ICD-9-CM: 428), or peripheral vascular disorders (ICD-9-CM: 440, 441.2, 441.4, 441.7, 441.9, 443, 444, and 447.1) with at least one hospitalization before the index date. All patients were followed from the index date to the date of aHV occurrence or death, withdrawal from the NHIRD, or 31 December 2013.

### 2.3. Statistical Analyses

Data are presented as the number (%) and mean (SD) for categorical and continuous variables, respectively. The differences for each variable in the SOT and non-SOT cohorts were standardized; a value of less than 0.1 indicates a negligible difference between the cohorts. The Kaplan–Meier method was applied to calculate the cumulative incidence curves of aHV in the groups, and the curves were compared using a log-rank test. The risks of aHV development and recurrence in the groups were assessed using crude and adjusted Cox proportional hazard models and are presented as hazard ratios (HRs) and adjusted HRs (aHRs) with 95% confidence intervals (CIs). In consideration of death events during the study period, a competing risk regression analysis was also performed. Statistical analyses were conducted with type I error α = 0.05 using a statistical software package, SAS version 9.4 (SAS Institute Inc., Cary, NC, USA).

## 3. Results

The study enrolled 18,064 individuals (Table 1): 9032 (50%) cases and 9032 controls without SOT. Approximately 60% of the population was male. The mean ages of each group were 48.7 and 49.5 years, respectively. By conducting propensity score matching according to age, sex, index year, and associated comorbidities, no significant differences were observed between the case and comparison groups for these factors (all standardized mean differences < 0.1).

Table 2 shows the risk factors for developing aHV infection. Patients with SOT (aHR = 9.19, 95% CI = 7.19–11.75, *p* < 0.001), female patients, those aged 50–64 years (aHR = 1.73, 95% CI = 1.47–2.03, *p* < 0.001), those aged older than 65 years (aHR = 1.47, 95% CI = 1.02–2.12, *p* = 0.038), and those with chronic kidney disease (aHR = 1.37, 95% CI = 1.13–1.66, *p* = 0.001) or heart failure (aHR = 1.39, 95% CI = 1.12–1.74, *p* = 0.003) had a significantly higher risk of developing aHV infection after adjustments for demographic factors and comorbidities. Stratified by age, gender, and associated comorbidity, the incidence rate of aHV infections was higher in patients with SOT (aHR = 9.19, 95% CI = 7.19–11.75) when compared with the general population. 

We evaluated the individual risk of VZV and HSV for SOT subgroups. SOT was categorized into four subgroups: kidney, heart, lung, and liver transplantation. The risks of VZV and HSV infections were examined. A significantly higher risk of VZV and HSV in kidney transplant recipients (aHR = 9.13, 95% CI = 6.88–12.12; aHR = 12.13, 95% CI = 6.46–22.77), heart transplant recipients (aHR = 14.34, 95% CI = 8.83–23.28; aHR = 18.54, 95% CI = 7.19–47.81), or liver transplant recipients (aHR = 5.90, 95% CI = 4.06–8.57; aHR = 8.28, 95% CI = 3.88–17.67) was observed after multivariable adjustment for demographic factors and comorbidities, respectively. Because of the limited sample size of lung transplants, we only observed a significant association between lung transplantation and VZV (aHR = 35.40, 95% CI = 15.48–80.96).

To assess the effect of age on the risk of aHV infection in SOT, we stratified three age groups into subgroups of kidney, heart, and liver (Table 3). The age groups were categorized as less than 50, 50–64, and older than 65 years, and they exhibited a significantly higher risk in all younger age groups in comparison to the non-SOT cohort, with the exception of heart transplant recipients (aHR = 2.50, 95% CI = 0.45–13.85) or liver transplant recipients (aHR = 2.37, 95% CI = 0.57–9.81) when older than 65 years.

Assessing the impact of post-transplant follow-up time on viral infection, the incidence of viral infection over follow-up time after transplantation was analyzed (Table 4). Patients with SOT had a 25.18-fold (aHR = 25.18, 95% CI = 13.36–47.43), 10.22-fold (aHR = 10.22, 95% CI = 6.36–16.41), and 4.89-fold (aHR = 4.89, 95% CI = 3.50–6.83) risk of developing aHV during a period after the transplant date within 1 year, between 1 and 3 year(s), and more than 3 years, respectively. The risk of virus infection over post-transplant follow-up time is plotted in Figure 1. 

We investigated the recurrence of aHV after transplantation (Table 5). The comparison shows that SOT was comparable to the non-SOT group in either overall recurrence risk (aHR = 1.75, 95% CI = 0.87–3.52) or risk within one year (aHR = 0.68, 95% CI = 0.35–1.33).

To evaluate the outcomes of SOT after virus infection, the death risk was determined in patients after aHV infection in both groups with or without SOT. A comparison of both groups shows that the risk of death (aHR = 0.68, 95% CI = 0.44–1.05) was not significant in SOT compared to non-SOT after adjustments for demographic factors and comorbidities.

## 4. Discussion

Solid organ transplant recipients are occasionally associated with an increased risk of cutaneous diseases. Compared to lethal skin cancers, virus infection is generally assumed to be more indolent. In the present study, which encompassed kidney, liver, heart, and lung organ recipients and included 9302 cases with persons of a young age over a 10-year observational period, a higher incidence rate of 12.5 × 10^3^ person-years developing severe alphaherpesvirus (aHV) was observed. The general population’s incidence rate in adults has been estimated to be 1.2–4.8 × 10^3^ [5,8]. A comparison of patients with or without transplants revealed that those who received a transplant had a nine-fold increased risk of severe aHV infection. This result was compatible with the previous study [7]. We calculated the subfamily herpes virus risk of developing HSV and VZV among four subgroups of organ recipients. Firstly, the risk of severe HSV infection remained increased, which was considered to be presumably mild in the general population. In HSV infection, multivariate regression analysis revealed a 12.1-fold increased risk in kidney transplant recipients, an 8.2-fold increased risk in liver recipients, and an 18.5-fold increased risk in heart recipients. To our knowledge, this is the largest study to disclose the epidemiological analysis of severe herpes simplex infection in SOT recipients.

Secondly, when stratifying by age, we observed that the risk of viral infection varied and was not parallel to an increase in age, a finding differing from those previously reported in the general population. Notably, the SOT recipient group aged younger than 49 years continued to exhibit a higher herpes virus risk, either HSV or VZV, when compared with the comparison group. These findings may suggest that reactivation of the herpes virus in organ recipients is independent of age. In immune-competent populations, the causal relationship between the virus and the natural senescence of cellular immunity interacts and presents an increasing risk with age. In transplant recipients, lifelong immunosuppressant treatment changes the relationship and leads to a left shift in the incidence curve to a younger age. Furthermore, our findings indicate that the majority of aHV infection cases occurred within one year post-transplant after a large dose of immunosuppressants given (at the early stage of transplantation), with about a 12-fold increased risk, and then the risk declined rapidly to 4-fold three years after transplantation.

Cell-mediated immunity has been suggested to play a more crucial role involving reactivation than humoral immunity; however, the mechanism underlying cellular and humoral immunity is still unclear [8,9,10,11]. Previous studies have suggested that the attenuated immunity of T cells is significantly correlated with disease severity instead of being about antibody level [8,9,10]. For example, several immunosuppressive conditions, such as HIV-positive patients, are also associated with increased herpes infections [12,13,14]. In HIV-positive patients, incidences ranging from 25.0 to 91.5 × 10^3^ person-years exhibiting 12–17-fold more significant risk of developing zoster have been reported. However, data for elucidating the influence of immunosuppressants on aHV reactivation in SOT recipients were limited mostly because of the small sample sizes.

A previous study reported that the incidence rate of VZV was the highest in the first year after transplantation, suggesting that acute T cellular immunity impairment occurs in the first several months following transplantation [15]. Third, with stratification by the type of organ transplant in VZV and HSV infection, multivariate regression analysis revealed a high virus risk in heart recipients. The lower VZV risk in liver transplant recipients was attributed to the use of relatively mild immunosuppressant agents because of the greater resistance to rejection in liver allografts [16]; in contrast, potent immunosuppressant agents in lung or heart recipients may account for the high risk of VZV reactivation [17]. Our findings support the results of previous trials and retrospective studies [8,12]. Altogether, our findings conclude that virus risk in organ recipients could be potentially modulated by the adjustment of immunosuppressants.

In organ recipients with herpes infection, antiviral agents (guanosine analogs) have been recommended for the universal treatment of aHV infections. The outcomes of the use of these agents in organ recipients with aHV infections and whether they are less effective in organ recipients under long-term immunosuppressive drug use have yet to be investigated. The risk of recurrence at one year and throughout the overall study follow-up period was examined in both groups, and our results found that it was not different between both groups. In addition, the death risk in patients after infection was also compared in both groups and was found to be comparable between groups (aHR = 0.68, 95% CI = 0.44–1.05) and not statistically different. These findings suggest that contemporary antiviral agents are effective in controlling herpes infection in organ recipients with long-term immunosuppressants. Whether the prophylactic use of anti-VZV agents is beneficial for organ recipients warrants investigation in the future [18].

These results are robust; nevertheless, limitations must be addressed. For the diagnosis of severe aHV infections, we adopted inpatient NHIRD data. The validation of aHV infection was dependent on patient records in hospital and is subject to peer review. In addition, under the universal health care system, the inpatients needed to receive antiviral medication or reimbursements for anti-aHV medication. Patients with severe infections such as extensively disseminated dermatome infection, hepatitis, pneumonitis, encephalitis, meningitis, Bell’s palsy, Ramsay–Hunt syndrome, and serious ophthalmologic trigeminal nerve involvements—caused by the reactivation of latent virus infection—were enrolled. Hence, the diagnosis of aHV infection was reliable. However, patients with fairly mild skin symptoms might be unavoidably excluded, and underestimation may have biased the results. In addition, laboratory data such as biochemistry profile, virus status, and personal lifestyle documentation are unavailable in the NHIRD. To overcome inherent potential confounds such as tobacco smoking and alcohol consumption habits—which can influence infection risk—proxy variables, including COPD incidence for smoking habits, hypertension, hyperlipidemia, and diabetic incidence of obesity, were used to reduce the effect of these confounders. Although we attempted to control for potential disease-associated confounds, unknown or unmeasured confounds might have biased our results.

Due to our exclusion criteria, data for younger patients less than 18 years old and the efficacy of vaccination were inadequate in the present study. In Taiwan, universal varicella vaccination was initially implemented in 2004 [19]. Of note, it has been previously reported that immunocompetent children who received the varicella vaccine were infected with herpes zoster later in childhood and adolescence [20,21]. Therefore, young patients with solid organ transplantation need to be more careful about this condition.

## 5. Conclusions

In the present study, we report a higher risk of severe alphaherpesvirus infection in solid organ transplant recipients than in the general population, particularly among younger recipients. In addition, to our knowledge, this is the largest study disclosing severe herpes simplex virus infections in solid organ recipients, which has often been easily ignored previously. We also report that the risk of aHV infection was highest in the first year after transplantation. Currently, available antiviral agents effectively control virus infection in organ recipients. To prevent progression to serious herpes infection, early treatment is recommended for organ recipients.

## Figures and Tables

**Figure 1 biomedicines-11-00637-f001:**
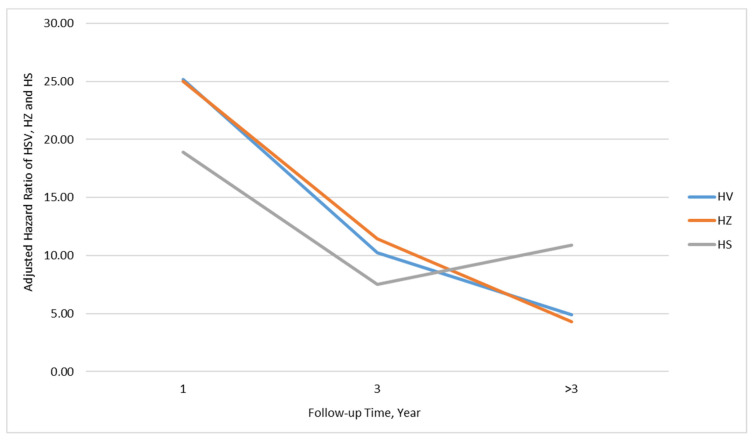
The incidence of developing aHZ in SOT recipients stratified by post-transplant period from an index date within 1 year, between 1 and 3 years, and after more than 3 years.

**Table 1 biomedicines-11-00637-t001:** Demographic characteristics and comorbidities of patients with new solid organ transplantation.

		Solid Organ Transplantation		Standardized Mean Differences ^§^
Variable	Total	No	Yes	
	(*n* = 18,064)	(*n* = 9032)	(*n* = 9032)	
	*n*	*n* (%)/Mean ± SD	*n* (%)/Mean ± SD	
Gender				
Female	6754	3390 (37.5)	3364 (37.2)	0.006
Male	11,310	5642 (62.5)	5668 (62.8)	0.006
Age at baseline				
<50	8869	4372 (48.4)	4497 (49.8)	0.028
50–64	8155	4080 (45.2)	4075 (45.1)	0.001
≥65	1038	578 (6.4)	460 (5.1)	0.057
Age, mean (SD)		49.5 (11.7)	48.7 (11.1)	0.069
Comorbidity				
Hypertension	9211	4646 (51.4)	4565 (50.5)	0.018
Hyperlipidemia	1892	968 (10.7)	924 (10.2)	0.016
Diabetes	3872	1972 (21.8)	1900 (21)	0.019
Cancer	2645	1360 (15.1)	1285 (14.2)	0.023
Chronic kidney disease	9964	4988 (55.2)	4976 (55.1)	0.003
COPD	402	203 (2.2)	199 (2.2)	0.003
Heart failure	2286	1122 (12.4)	1164 (12.9)	0.014
Peripheral vascular disorders	356	158 (1.7)	198 (2.2)	0.032

Abbreviation: COPD, chronic obstructive pulmonary disease. ^§^: A standardized mean difference of ≤0.1 indicates a negligible difference between the two cohorts. Reference from [7].

**Table 2 biomedicines-11-00637-t002:** Patients’ risk of herpes virus infection with and without solid organ transplantation.

Characteristics	Event	Crude		Adjusted	
	(*n* = 676)	HR (95% CI)	*p* Value	HR (95% CI)	*p* Value
Solid Organ Transplantation					
No	71	Ref.		Ref.	
Yes	605	9.04 (7.07–11.56)	<0.001	9.19 (7.19–11.75)	<0.001
Gender					
Female	340	Ref.		Ref.	
Male	336	0.65 (0.56–0.76)	<0.001	0.65 (0.56–0.76)	<0.001
Age at baseline					
<50	290	Ref.		Ref.	
50–64	352	1.60 (1.37–1.87)	<0.001	1.73 (1.47–2.03)	<0.001
≥65	34	1.33 (0.93–1.89)	0.121	1.47 (1.02–2.12)	0.038
Baseline comorbidity					
Hypertension	411	1.47 (1.26–1.71)	<0.001	1.18 (0.99–1.40)	0.060
Hyperlipidemia	73	1.15 (0.90–1.47)	0.251	0.96 (0.74–1.23)	0.732
Diabetes	150	1.28 (1.07–1.54)	0.007	1.12 (0.92–1.36)	0.260
Cancer	54	0.67 (0.51–0.89)	0.006	0.81 (0.60–1.08)	0.153
Chronic kidney disease	467	1.51 (1.28–1.78)	<0.001	1.37 (1.13–1.66)	0.001
COPD	18	1.49 (0.93–2.39)	0.093	1.31 (0.81–2.12)	0.267
Heart failure	102	1.38 (1.12–1.70)	0.003	1.39 (1.12–1.74)	0.003
Peripheral vascular disorders	21	1.85 (1.20–2.85)	0.006	1.35 (0.87–2.09)	0.183

Abbreviation: HR, hazard ratio; CI, confidence interval. Adjusted HR: adjusted for age; all comorbidities in Cox proportional hazards regression. Herpes virus infection (ICD-9: 053, 054). Reference from [7].

**Table 3 biomedicines-11-00637-t003:** aHV infection risk in subgroups of organs by stratification of age.

Variables	Non-SOT			SOT			SOT versus Non-SOT	
	*n* = 9032			*n* = 9032			Crude HR	Adjusted HR
	Event	Person Years	IR	Event	Person Years	IR	(95% CI)	(95% CI)
Kidney type								
<50 years	29	28,545	1.02	212	21,036	10.08	10.20 (6.92–15.04) ***	9.28 (6.23–13.83) ***
50–64	33	20,342	1.62	203	10,563	19.22	12.56 (8.69–18.15) ***	10.28 (6.97–15.18) ***
≥65	9	2396	3.76	18	1066	16.89	4.66 (2.08–10.42) ***	5.02 (2.02–12.47) ***
Heart type								
<50 years	29	28,545	1.02	21	1710	12.28	11.91 (6.78–20.93) ***	19.22 (6.67–55.41) ***
50–64	33	20,342	1.62	39	1438	27.12	16.70 (10.50–26.55) ***	19.08 (7.97–45.67) ***
≥65	9	2396	3.76	3	203	14.81	3.92 (1.06–14.53) *	2.50 (0.45–13.85)
Liver type								
<50 years	29	28,545	1.02	24	4866	4.93	4.66 (2.70–8.04) ***	9.44 (4.16–21.42) ***
50–64	33	20,342	1.62	74	6453	11.47	6.55 (4.33–9.90) ***	8.45 (5.13–13.90) ***
≥65	9	2396	3.76	4	690	5.80	1.58 (0.48–5.17)	2.37 (0.57–9.81)

*: *p*-value < 0.05; ***: *p*-value < 0.001.

**Table 4 biomedicines-11-00637-t004:** Incidence and hazard ratio of aHV stratified by follow-up year.

Variables	Non-SOT			SOT			SOT versus Non-SOT	
	*n* = 9032			*n* = 9032			Crude HR	Adjusted HR
	Event	Person Years	IR	Event	Person Years	IR	(95% CI)	(95% CI)
Follow-up Year								
<1	10	8825	1.13	235	8403	27.97	24.53 (13.03–46.20) ***	25.18 (13.36–47.43) ***
1–3	19	14,969	1.27	177	13,905	12.73	10.03 (6.25–16.09) ***	10.22 (6.36–16.41) ***
>3	42	27,489	1.53	193	25,881	7.46	4.89 (3.51–6.83) ***	4.89 (3.50–6.83) ***

Abbreviation: IR, incidence rates per 1000 person-years; HR, hazard ratio; CI, confidence interval. Adjusted HR: adjusted for gender, age, and all comorbidities in Cox proportional hazards regression. ***: *p*-value < 0.001.

**Table 5 biomedicines-11-00637-t005:** Recurrent risk of aHV infection in groups through competing-risks regression.

Variable	Solid Organ Transplantation		*p*-Value
	No	Yes	
aHV Recurrence (*n* = 101)			
Crude HR (95% CI)	1.00 (Ref.)	1.78 (0.90–3.54)	0.10
Adjusted HR ^†^ (95% CI)	1.00 (Ref.)	1.75 (0.87–3.52)	0.12
1-year aHV Recurrence (*n* = 83)			
Crude HR (95% CI)	1.00 (Ref.)	0.76 (0.40–1.43)	0.39
Adjusted HR ^†^ (95% CI)	1.00 (Ref.)	0.68 (0.35–1.33)	0.26

Adjusted HR ^†^: multivariable analysis including all factors in the univariable cox model.

## Data Availability

The datasets supporting the conclusions of this article are managed by Taiwan’s Ministry of Health and Welfare (MOHW). The MOHW approved our application to access these data. Any investigator interested in accessing this dataset must submit an application to the MOHW. The MOHW’s address is No. 488, Sec. 6, Zhongxiao E. Rd., Nangang Dist., Taipei City 115, Taiwan (R.O.C.); Phone: +886-2-8590-6848. Please contact MOHW personnel (email: stcarolwu@mohw.gov.tw) for further assistance. All relevant data are provided in this paper.

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
