# Peer review of "Risk of Severe Alphaherpesvirus Infection after Solid Organ Transplantation: A Nationwide Population-Based Cohort Study"

_biomedicines, 2023, doi:10.3390/biomedicines11020637_

Round 1
Reviewer 1 Report
The Taiwan National Health Insurance Research Database has become a valuable resource for epidemiological investigations. In this manuscript, the authors have assessed the number of alpha herpesvirus infections in patients who have received solid organ transplants. The data will be of interest to many medical centers where transplantation is performed. A few comments for improvement are listed below.
1. Abstract. State in the Abstract that the transplants included heart, lung, liver and kidney.
2. Methods, Study population, line 80. The authors never tell us if the protocols for management of patients with solid organ transplants include routine administration of intravenous or oral acyclovir or any other antiviral medications. Please add this information into this section of Methods. The authors may not be able to determine whether each patient did or did not receive acyclovir, but the authors should be able to find out whether most post-transplantation protocols included administration of antiviral medications in hospitals in Taiwan between 2000-2012.
3. Results, line 111. Give the range in age of subjects enrolled in the transplant group and the control group. If some subjects were age 18 or younger, state how many subjects were in this group of children and adolescents. This information should be written into the text of Results.
4. Results, lines 138-143. The authors state that there was a higher risk of HSV/VZV reactivation in all younger groups. See comment 3. Again, describe in the text what were the ages of the youngest transplant patients who had increased rates of HSV/VZV.
5. Discussion, line 224. What is missing from Discussion is a mention of varicella vaccination in Taiwan. Universal varicella vaccination was initiated in 2004. See article by H. Cheng et al, Epidemiology of breakthrough varicella after implementation of universal varicella vaccination program in Taiwan 2004-2014, Scientific Reports 8:17192, 2018.
Furthermore, children who are given varicella vaccination can present with herpes zoster caused by the vaccine virus later in childhood or in adulthood. Please read the two following papers. (a) R. Harpaz et al, The epidemiology of herpes zoster of varicella and herpes zoster vaccines. Clin Infect Dis. 69:345, 2019. (b) A. Moodley et al, Severe herpes zoster following varicella vaccination in immunocompetent young children. J. Child Neurology 34:184, 2019.
Please add a short paragraph to the Discussion to alert medical professionals in Taiwan and elsewhere that some of their cases of herpes zoster in younger solid organ transplant patients could be caused by reactivation of the varicella vaccine virus.
6. References. Add the above 3 papers into the list of references.
Author Response
Response to Reviewer 1 Comments
Point 1: Abstract. State in the Abstract that the transplants included heart, lung, liver and kidney.
Response 1: Thank you for the comment. It had been added in the abstract on line 22.
Point 2: Methods, Study population, line 80. The authors never tell us if the protocols for management of patients with solid organ transplants include routine administration of intravenous or oral acyclovir or any other antiviral medications. Please add this information into this section of Methods. The authors may not be able to determine whether each patient did or did not receive acyclovir, but the authors should be able to find out whether most post-transplantation protocols included administration of antiviral medications in hospitals in Taiwan between 2000-2012.
Response 2: Thank you for the comment. It had been added on line 98.
“The protocols for managing patients with solid organ transplants caused by aHV infections included routinely administering intravenous or oral acyclovir or other antiviral medications.”
Point 3: Results, line 111. Give the range in age of subjects enrolled in the transplant group and the control group. If some subjects were age 18 or younger, state how many subjects were in this group of children and adolescents. This information should be written into the text of Results.
Response 3: Thank you for the comment. “All patients included were older than 18 years old.” It had been added on line 97.
Point 4: Results, lines 138-143. The authors state that there was a higher risk of HSV/VZV reactivation in all younger groups. See comment 3. Again, describe in the text what were the ages of the youngest transplant patients who had increased rates of HSV/VZV.
Response 4: Thank you for the comment. It had been added on line 97. “All patients included were older than 18 years old.”
Point 5: Discussion, line 224. What is missing from Discussion is a mention of varicella vaccination in Taiwan. Universal varicella vaccination was initiated in 2004. See article by H. Cheng et al, Epidemiology of breakthrough varicella after implementation of universal varicella vaccination program in Taiwan 2004-2014, Scientific Reports 8:17192, 2018.
Furthermore, children who are given varicella vaccination can present with herpes zoster caused by the vaccine virus later in childhood or in adulthood. Please read the two following papers. (a) R. Harpaz et al, The epidemiology of herpes zoster of varicella and herpes zoster vaccines. Clin Infect Dis. 69:345, 2019. (b) A. Moodley et al, Severe herpes zoster following varicella vaccination in immunocompetent young children. J. Child Neurology 34:184, 2019.
Please add a short paragraph to the Discussion to alert medical professionals in Taiwan and elsewhere that some of their cases of herpes zoster in younger solid organ transplant patients could be caused by reactivation of the varicella vaccine virus.
Response 5: Thank you for the comment. It had been added on line 376.
“Due to exclusion criteria, data for elucidating the younger patients less than 18 years old and the efficacy of vaccination were inadequate in the present study. In Taiwan, universal varicella vaccination was implemented initially in 2004 [18]. Of note, it has been previously reported that immunocompetent children who received the varicella vaccine were infected with herpes zoster later in childhood and adolescence [19, 20]. Therefore, young patients with solid organ transplantation need to be more careful about this condition.”
.
Point 6: References. Add the above 3 papers into the list of references.
Response 6: Thank you for the comment. The above three papers had been added into the list of references
Reviewer 2 Report
This paper is very interesting but much improvement is needed.
1) What concerns me most is the syntac and expressions used in this paper. This is however understandable as English is not the first language of the authors. The paper needs to be edited ny an English-language editor who can look out for the correct expression of the sentences.
2) On many occasions, the sentences failed to use the correct superlative terms when necessary. There is a big difference in the meanings of "high" and "higehr", For example in the abstract, " A high risk of alphaherpes virus infection", what the statistics is showing is HIGHER risk, not necessrily high risk. If you want to claim high risk then you must show the absolute chances of infection.. Such mistake is repeated throughout the paper including line 181. Proper usage of the English language is absolutely necessary to convey the correct scientific meaning.
3) The sentence in the abstract, "To investigate the risk of alphaherpesvirus infection, including herpes 21simplex and herpes zoster after SOT" is not complete.We cannot write incomplete sentences
4) On several occasions eg LINE 221, there are paragraphs with just one sentences. This is very awkward and I don't know if the paper has been misaligned or it is a mistake.
5) Line 49 "Board" should be changed to "Broad"
6) LINE 246 "Sever" needs to be corrected.
7) Scientifically, I don't understand why there should be such infections from transplants in the first place since antibody tests are available for the viruses. I know that tests for HIV is conducted before transplant but why not for these viruses? Such should be addressed in the paper along with with the necessary references.
8) Since the authors are from well-known Universities in Taiwan, one idea, in addition to the extensive editing by the journal's English-language eidtor, would be for the authors to approach someone from their University's English language department, who can help them with the correct English expression via Chinese-English translation. Or collaborate with with someone in the USA, UK or former British colones like Singapore or Hong Kong
Author Response
Response to Reviewer 2 Comments
This paper is very interesting but much improvement is needed.
Point 1: What concerns me most is the syntac and expressions used in this paper. This is however understandable as English is not the first language of the authors. The paper needs to be edited by an English-language editor who can look out for the correct expression of the sentences.
Response 1: Thank you for the comment. It had been edited by an English-language editor again.
Point 2: On many occasions, the sentences failed to use the correct superlative terms when necessary. There is a big difference in the meanings of "high" and "higehr". For example, in the abstract, " A high risk of alphaherpes virus infection", what the statistics is showing is HIGHER risk, not necessarily high risk. If you want to claim high risk then you must show the absolute chances of infection. Such mistake is repeated throughout the paper including line 181. Proper usage of the English language is absolutely necessary to convey the correct scientific meaning.
Response 2: Thank you for the comment. It had been corrected.
Point 3: The sentence in the abstract, "To investigate the risk of alphaherpesvirus infection, including herpes 21simplex and herpes zoster after SOT" is not complete. We cannot write incomplete sentences.
Response 3: Thank you for the comment. It had been corrected on line 19.
“The aim of our study was to investigate the risk of alphaherpesvirus infections, including herpes simplex and herpes zoster, after solid organ transplantation.”
Point 4: On several occasions eg. LINE 221, there are paragraphs with just one sentences. This is very awkward and I don't know if the paper has been misaligned or it is a mistake.
Response 4: Thank you for the comment. It had been corrected
Point 5: Line 49 "Board" should be changed to "Broad"
Response 5: Thank you for the comment. It had been corrected on line 56
Point 6: LINE 246 "Sever" needs to be corrected.
Response 6: Thank you for the comment. It had been corrected on line 387.
Point 7: Scientifically, I don't understand why there should be such infections from transplants in the first place since antibody tests are available for the viruses. I know that tests for HIV is conducted before transplant but why not for these viruses? Such should be addressed in the paper along with the necessary references.
Response 7: Thank you for the comment. It had been added on line 69.
“A herpes virus test is examined routinely before solid organ transplantation. Patients who have previously been infected with the herpes virus are not barred from receiving organ transplants, so they can receive the transplant regardless of their herpes infection history.”
Point 8: Since the authors are from well-known Universities in Taiwan, one idea, in addition to the extensive editing by the journal's English-language eidtor, would be for the authors to approach someone from their University's English language department, who can help them with the correct English expression via Chinese-English translation. Or collaborate with with someone in the USA, UK or former British colones like Singapore or Hong Kong
Response 8: Thank you for the comment. It had been edited by an English-language editor again.
Round 2
Reviewer 2 Report
Improvements seen